# Enhancing Influenza Epidemics Forecasting Accuracy in China with Both Official and Unofficial Online News Articles, 2019–2020

**DOI:** 10.3390/ijerph18126591

**Published:** 2021-06-18

**Authors:** Jingwei Li, Choon-Ling Sia, Zhuo Chen, Wei Huang

**Affiliations:** 1School of Management, Xi’an Jiaotong University, Xi’an 710049, China; li.jing.wei123@stu.xjtu.edu.cn; 2Department of Information Systems, City University of Hong Kong, Hong Kong 999077, China; iscl@cityu.edu.hk; 3College of Public Health, University of Georgia, Athens, GA 30602, USA; zchen1@uga.edu; 4School of Economics, University of Nottingham Ningbo China, Ningbo 315000, China; 5College of Business, Southern University of Science and Technology, Shenzhen 518000, China

**Keywords:** digital disease detection, seasonal influenza surveillance, online news, autoregressive exogenous model

## Abstract

Real-time online data sources have contributed to timely and accurate forecasting of influenza activities while also suffered from instability and linguistic noise. Few previous studies have focused on unofficial online news articles, which are abundant in their numbers, rich in information, and relatively low in noise. This study examined whether monitoring both official and unofficial online news articles can improve influenza activity forecasting accuracy during influenza outbreaks. Data were retrieved from a Chinese commercial online platform and the website of the Chinese National Influenza Center. We modeled weekly fractions of influenza-related online news articles and compared them against weekly influenza-like illness (ILI) rates using autoregression analyses. We retrieved 153,958,695 and 149,822,871 online news articles focusing on the south and north of mainland China separately from 6 October 2019 to 17 May 2020. Our model based on online news articles could significantly improve the forecasting accuracy, compared to other influenza surveillance models based on historical ILI rates (*p* = 0.002 in the south; *p* = 0.000 in the north) or adding microblog data as an exogenous input (*p* = 0.029 in the south; *p* = 0.000 in the north). Our finding also showed that influenza forecasting based on online news articles could be 1–2 weeks ahead of official ILI surveillance reports. The results revealed that monitoring online news articles could supplement traditional influenza surveillance systems, improve resource allocation, and offer models for surveillance of other emerging diseases.

## 1. Introduction 

Influenza is a severe public health concern internationally. Every year, influenza is estimated to cause 1 billion cases, 3–5 million severe cases, and 290,000–650,000 influenza-related respiratory deaths worldwide [1,2] and cause an estimated 88,100 influenza-related excess respiratory deaths in China [3]. Through monitoring and evaluating influenza transmission, real-time influenza surveillance can contribute to early response and preparation for influenza epidemics [4]. The accuracy of real-time influenza activity forecasting strongly influences the effectiveness of the response and preparation [5]. The commonly used and well-established method to track influenza activity is through influenza-like illness (ILI) reporting from sentinel hospitals all over the country [4,6]. However, the official ILI reports suffer from a delay of one to three weeks due to processing ILI-related information from sentinel hospitals [5,7]. Traditional influenza epidemics forecasting methods based on historical ILI reports cannot capture the influenza activity in a timely manner and sometimes suffer from overfitting, especially during influenza outbreaks [5,8]. Thus, it is essential to seek real-time novel data sources to enhance influenza activity forecasting accuracy.

Public online data sources, including search queries, social media, and online news articles, accumulate a large volume of data every day and have shown a great potential to supplement traditional infection disease surveillance [9,10,11]. Specifically, public online data sources could improve the timelines and forecasting accuracy for infectious disease surveillance, compared with traditional official reporting results [12,13]. Public online data contains a large amount of unstructured data and a combination of methods has been applied to support the analysis. These methods include text mining, machine learning, and regression analysis [8,10,14]. Much influential infection disease surveillance has benefitted from using public online data. For example, Wilson and Brownstein (2009) used online articles to support timely surveillance of Listeria [15], He et al. (2018) made use of the Baidu search index to predict the incidence of HIV in China [16], and Liu et al. (2020) incorporated both search query data and online news articles data to provide real-time forecasting of COVID-19 outbreaks in different Chinese provinces [17].

Influenza is one of the most impactful infection diseases worldwide [1]. Digital influenza surveillance, which incorporates various kinds of real-time influenza-related information from the Internet, has been a valuable complement to the traditional influenza surveillance method based on historical ILI reports [6]. Previous studies made use of influenza-related online information, including the fraction of Google search queries [5,7], the number of tweets or microblogs [8,18], Wikipedia article views [19], Flu Near You [20], and self-reporting digital applications [6]. Among them, search queries and social media like Twitter received the most attention and accounted for 95.1% of internet-based health surveillance studies from a recent literature review [21]. However, methods based on Google search queries suffered an instability problem during the influenza outbreak [22], while methods based on tweets suffered from noise with linguistic errors and idiosyncratic style [8,23]. Online news articles are richer in information and written in a more formal language, compared with search queries and tweets. The number of online news articles has increased dramatically recently for the incorporation of new, emerging unofficial user-generated news into traditionally officially-certificated organization-generated news [24]. However, the value of online news articles in disease surveillance has not been fully studied. Previous studies only focused on the news articles that mention the “outbreak” of disease like Healthmap [25,26] or used the news articles from official news websites only [27,28].

In this study, we propose a novel approach to estimate influenza activity in both the south and north of mainland China by adding information extracted from both official and unofficial online news articles as a new exogenous input into the traditional auto-regression model. The provinces in south of mainland China include Shanghai, Jiangsu, Zhejiang, Anhui, Fujian, Jiangxi, Hubei, Hunan, Guangdong, Guangxi, Hainan, Chongqing, Sichuan, Guizhou, and Yunnan, and provinces in the north of mainland China include Beijing, Tianjin, Hebei, Shanxi, Inner Mongolia, Liaoning, Jilin, Heilongjiang, Shandong, Henan, Tibet, Shaanxi, Gansu, Qinghai, Ningxia, and Xinjiang. Our study used data from hundreds of millions of influenza-related online news articles in Chinese media. It tested whether incorporating online news article data can improve forecasting accuracy during influenza outbreaks. Our proposed model was compared with baseline models based on other online data sources. 

The rest of this paper is organized as follows. First, we describe datasets used in this study, including influenza-related online news articles and influenza-related microblogs from a Chinese commercial online platform and influenza-like illness rates from the Chinese National Influenza Center (CNIC). A data mining method to extract influenza-related online news articles is also used in this part. We then present the descriptive analysis for the data we retrieved in the previous part. The fractions of influenza-related online news articles and microblogs are compared with the official ILI rates to characterize their temporal distribution during the seasonal influenza epidemics from 2019 to 2020. Further, autoregression analysis based on different online data sources is conducted to compare the influenza epidemics forecasting accuracy between them over time. Moreover, paired t-tests are used to determine whether there is a statistically significant improvement in the forecasting accuracy comparing our proposed model to other baseline models.

## 2. Materials and Methods

### 2.1. Data Collection and Processing

Social media data used in this study, including online news articles and microblogs, are retrieved from the Sina Network Opinion Surveillance System (SNOSS) (https://www.yqt365.com, accessed on 15 October 2020). SNOSS is a commercially available online platform that continuously crawls all kinds of public online data in China. The platform crawls more than 140 million pieces of data every day, consisting of microblogs (Chinese Twitter), online news articles, announcements from Chinese Government websites, etc. Online news articles are retrieved from officially certificated sources, including People’s Daily, and unofficial sources, e.g., user-generated news in Wechat Official Account [24]. Data in SNOSS contains tags for time and location, representing when and where the data are posted. However, different from microblogs, where users usually post about events near themselves [29], online news articles can post news about incidents in other places. To extract online news articles that covered certain localities, we used an additional location filtering strategy. Specifically, we identified a news article covering a certain location when the place’s name appeared to be the highest frequency of occurrence in the news article compared with other places’ names. Our study used the 33 provinces’ names to extract news articles covering the south and north of mainland China separately. The primary aim of SNOSS is to monitor real-time hot topics online, including breaking news and public concerns, which also shows a great potential to improve influenza surveillance through real-time online data [7].

#### 2.1.1. Influenza-Related Online News Articles

We collected influenza-related online news articles from SNOSS from 6 October (week 40) 2019 to 17 May (week 20) 2020. We collected data from this timeframe based on World Health Organization’s guidance. The guidance stated that temperate climate zones in the northern hemisphere (including mainland China) had clear seasonality and a well-defined flu season, which started from week 40 of one year to week 20 of the following year [30]. This well-defined flu season timeframe is used as the influenza surveillance period for other northern hemisphere temperate countries or organizations, including America [31], England [32], and the European Union [33], and this timeframe is also used by related influenza forecasting research [5,7].

We extracted online news articles that focus on influenza by using a combination of two sets of keywords [27]. One set of keywords includes ‘flu’ and ‘influenza’, and the other set contains keywords mostly related to different aspects of the flu. To identify keywords from online news articles that are most relevant to flu, we first used the bag-of-words method by treating all the online news articles with ‘flu/influenza’ as a document and identified 200 keywords with the highest frequency. We then used the TF-IDF method [34] to identify keywords that are low in frequency but represent a unique aspect of influenza. Developed by Salton and Buckley [35], the TF-IDF method is able to calculate the importance of a particular word in a document and provide more accurate representation and clustering results (see Appendix A, for detailed formula). Specifically, we built a term-document matrix based on 1000 randomly selected influenza-related articles and identified the top 100 terms (keywords) with the highest correlation with the word ‘influenza’ or ‘flu’. Finally, we invited a medical doctor to assist in selecting exact influenza-related keywords from the pool of words identified in the previous two steps. The glossary of influenza (flu) terms from the CDC (https://www.cdc.gov/flu/about/glossary.htm, accessed on 20 October 2020) is also considered in the final selection process. Finally, we identified 91 keywords mostly related to different aspects of influenza from the text mining process (see Appendix A).

To further filter out the noise in online news articles, we adopted two information filtering methods, which exclude news articles with specific keywords and exclude similar news articles with an 85% repetition rate within 35 days. Following Kim and Ahn [27], we filtered out news articles containing the keyword ‘avian influenza’, which has a small number of human contact cases and has not observed sustained human-to-human transmission [36]. We also filtered out news articles containing the keyword ‘novel coronavirus’, which could refer to a new emerging coronavirus-related disease like SARS or COVID-19. Then, we filtered out the republished online news articles by excluding similar articles with an 85% repetition rate within 35 days. There is a tendency to republish hot news articles to attract public attention. Filtering out republished online news articles can reduce this noise, which has caused estimation instability in Google Flu Trend [22]. 

Finally, we collected the number of weekly influenza-related online news articles and overall crawled online news articles from the south and north of mainland China separately. The fraction of weekly influenza-related online news articles compared to overall crawled online news articles was computed for the south and north mainland China separately and used as an independent variable in the models.

#### 2.1.2. Influenza-Related Microblogs

We also collected microblogs, which have been widely used in influenza surveillance with social media [37,38,39], for comparison. We followed the semantic filtering method proposed by Doan et al. [39]. We first extracted the microblogs containing at least one of 37 influenza-related symptom keywords or the keywords of ‘flu’ or ‘influenza’ (see Appendix A). We then filtered out the retweeted microblogs and microblogs with smiley emoticons, humor features, or URLs (see Appendix A). Finally, we collected the weekly number of influenza-related microblogs and overall crawled microblogs from the south and north of mainland China. We collected the data from online news articles and microblogs weekly and from the south and north mainland China separately to match the official influenza-like illness data, which is our baseline and will be introduced in the next section. The fraction of weekly influenza-related microblogs compared to overall crawled microblogs was then computed for the south and north mainland China separately and used as an independent variable in the models.

#### 2.1.3. Influenza-Like Illness Rates in China

Following the period of the online data sources we gathered, we collected weekly Influenza-like illness (ILI) rates for both the south and north of mainland China separately from the CNIC from 6 October 2019 to 17 May 2020. Based on guidelines from CNIC (http://ivdc.chinacdc.cn/cnic/, accessed on 15 October 2020), ILI case counts are defined as having a temperature of 38 °C or greater and having symptoms of cough or sore throat. Sentinel hospitals upload ILI case counts and total physician visits data to CNIC’s system on Monday of the following week. The CNIC would release the aggregated data at the end of next week. Due to the vast difference in climate and space, the CNIC reports the ILI rates for the south and north of mainland China separately. The data from the CNIC only includes seasonal influenza, while avian flu (H7N9) and swine flu (H1N1) are not included.

### 2.2. Statistical Analysis

#### 2.2.1. Descriptive Analysis

We summarized the number of influenza-related and overall data of online news articles and microblogs. The percentages of official and unofficial online news articles were also presented. We then compared the fraction of online news articles and microblogs with the ILI data to depict the data’s tendency and show their potential to provide earlier warnings for influenza outbreaks.

#### 2.2.2. Model Formulation

Similar to previous influenza surveillance research [5,6,8] that incorporates data sources into official ILI rates, we used a model in the form of logistic auto-regression with exogenous inputs [40]. Let $y_{t}=logit\left(p_{t}\right)$ be the logit-transformed CNIC’s ILI rates $p_{t}$ at week *t* and $x_{t}$ be the log-transformed fraction of influenza-related online news articles at week *t*. Our proposed model incorporated the online news articles as a new exogenous input into the auto-regression model:(1)$$y_{t}=\sum_{i=1}^{Lag_{ILI}}a_{i}y_{t-i}+\sum_{j=0}^{Lag_{News}}b_{j}x_{t-j}+c+\varepsilon_{t},\;\varepsilon_{t}\;\sim \;N\left(0,\;\sigma^{2}\right)$$
where $b_{j}$ quantifies the contribution from the fraction of online news articles at time t−j to predict CNIC’s ILI rates at time t, in addition to the historical CNIC’s ILI rates’ contribution. In this model, c is a constant term, $\varepsilon_{t}$ is a sequence of independent random variables, and $Lag_{ILI}$ and $Lag_{News}$ are the lagged indicators. $Lag_{ILI}\;$and $Lag_{News}$ are unknown and need to be estimated to achieve the highest prediction accuracy. Following previous research settings [5,6], we considered three weeks’ time lag and varied$\;Lag_{ILI}\;$from 1 to 3 and $Lag_{News}$ from 0 to 3. We then denoted our model as AR($Lag_{ILI}$) + News($Lag_{News}$), in which $Lag_{ILI}$ and $Lag_{News}$ were the optimal values to maximize prediction accuracy.

#### 2.2.3. Parameters Estimation and Baseline Models Comparison

Retrospective estimations of weekly ILI rates for the south and north of mainland China were produced through our proposed model, from 6 October 2019 to 17 May 2020. By calculating a collection of accuracy measures, we compared our proposed model’s estimates with the actual value, i.e., the CNIC’s weekly reported ILI rates for the south and north of mainland China, published with a delay of one to two weeks. The accuracy measures include the root-mean-squared error (RMSE), coefficient of determination (R^2^), mean absolute error (MAE), correlation with forecasting target, and correlation of increment with forecasting target (accuracy indexes are described in Appendix B). The same accuracy indexes were also calculated for other baseline models we built for comparison. The baseline models include:Autoregression model based on official ILI rates only [22,41], denoted as AR($lag_{ILI1}$),
(2)$$y_{t}=\sum_{i=1}^{Lag_{ILI1}}a_{i}y_{t-i}+c+\varepsilon_{t},\;\varepsilon_{t}\;\sim \;N\left(0,\;\sigma^{2}\right)$$Autoregression model based on official ILI rates and fraction of flu-related microblogs [8], denoted as AR($lag_{ILI2}$) + Mblog($lag_{Mblog2}$),
(3)$$y_{t}=\sum_{i=1}^{Lag_{ILI2}}a_{i}y_{t-i}+\sum_{h=0}^{Lag_{Mblog2}}d_{h}z_{t-h}+c+\varepsilon_{t},\;\varepsilon_{t}\;\sim \;N\left(0,\;\sigma^{2}\right)$$Autoregression model based on official ILI rates, fraction of flu-related online news articles, and fraction of present week’s flu-related microblogs [26,41], denoted as AR($lag_{ILI3}$) + News($lag_{News3}$) + Mblog(0),
(4)$$y_{t}=\sum_{i=1}^{Lag_{ILI3}}a_{i}y_{t-i}+\sum_{j=0}^{Lag_{News3}}b_{j}x_{t-j}+d_{t}z_{t}+c+\varepsilon_{t},\;\varepsilon_{t}\;\sim \;N\left(0,\;\sigma^{2}\right)$$
where $z_{t}$ is the log-transformed fraction of influenza-related microblogs at week *t*. 

All the time lags represent the number of previous weeks’ data that we considered in the model. Mblog(0) means that only the present week’s microblog was added to our proposed model, as suggested in [25,38]. $lag_{ILI1}$, $lag_{ILI2}$, and $lag_{ILI3}$ vary from 1 to 3, while $lag_{Mblog2}$ and $lag_{News3}$ range from 0 to 3. The values of these different time lags were determined by choosing the value leading to the lowest RMSE for each model [8,42] (see Appendix A).

We used the repeated k-fold cross-validation approach [43] to validate our proposed model and all the baseline models for fair comparisons. In k-fold cross-validation, the dataset is divided into k (approximately) equally sized subsets. In each interaction, one subset is used as the validation data, and the remaining (k − 1) subsets are used as training data. The process is repeated k times until each subset is treated as validation data once. We then repeated the cross-validation procedure multiple times by splitting the data into k different subsets. In this study, we set k = 10 and repeated the cross-validation procedure ten times, as suggested in [26]. We tested the hypothesis at a significance level of 0.05 and conducted all the analyses with the R version 4.0.2 statistical software packages DAAG [44] version 1.24 and caret [45] version 6.0-86.

## 3. Results

Overall, we retrieved 153,958,695 online news articles focusing on the south of mainland China and 149,822,871 online news articles focusing on the north of mainland China, from 6 October 2019 to 17 May 2020, from the SNOSS. In the south, 10,851,588 online news articles were from official sources, which accounted for 7.0% of the overall online news articles. In the north, 10,556,304 online news articles were from officially certificated sources, accounting for 7.0% of the overall online news articles. SNOSS also averagely traced 54,572,133 microblogs from the south and 72,182,715 microblogs from the north. The weekly time series of CNIC’s ILI rates, the fraction of influenza-related online news articles, and the fraction of influenza-related microblogs are displayed in Figure 1 and Figure 2. In both the south and north of mainland China, the highest official ILI rate occurred on 5 February 2020, while the highest fractions for influenza-related online news articles and microblogs occurred on 26 January 2020, which was one week earlier compared to the official report.

In Table 1 and Table 2, the optimal lags for different data sources, leading to the lowest RMSE for each model, are shown in the parentheses. Boldface highlights the best performance for each index in south and north mainland China. The paired *t*-test results, comparing our proposed model with baseline models, are shown in the last column. The *t* value is present in the last column with the related *p*-value in the parentheses below it. *** means significant at 1% level, ** means significant at 5% level, and * means significant at 10% level.

Table 1 and Table 2 summarize the accuracy indexes for all the forecasting methods in the south and north of mainland China separately, from 6 October 2019 to 17 May 2020. To test whether our proposed model can significantly improve the forecasting accuracy compared to other baseline models, we conducted paired *t*-tests on the RMSE of different models [46]. The paired *t*-test results are also shown in Table 1 and Table 2. Plots comparing different models’ forecasting results for south and north mainland China are shown in Figure 3 and Figure 4.

Results in Table 1 and Table 2 and Figure 3 and Figure 4 show that our proposed model incorporating online news articles data (AR + News) could improve the forecasting accuracy in terms of all accuracy indexes in both the south and north of mainland China. This result outperformed the baseline models using other data sources, including using previous official ILI data (AR) or incorporating microblog data (AR + Mblog). When our proposed model incorporates the present week’s microblog data (AR + News + Mblog), the accuracy declined in terms of RMSE, R^2^, and MAE, compared to the proposed model (AR + News). In terms of correlations and incremental correlations in the south and north of mainland China, AR + News (*r* = 0.98, incremental *r* = 0.824 in the south; *r* = 0.978, incremental *r* = 0.834 in the north) has similar performance to AR + News + Mblog (*r* = 0.985, incremental *r* = 0.839 in the south; *r* = 0.98, incremental *r* = 0.82 in the north). Close inspection in the south suggests that adding microblogs as exogenous input (AR + Mblog) can decrease the forecasting accuracy compared to the traditional autoregression model (AR) in terms of RMSE, R2, and MAE. Referring to RMSE, the accuracy decrease was statistically significant (*t* = −1.762, *p* = 0.081).

To assess the statistical significance of the improved prediction power, we conducted a paired *t*-test on RMSE to compare our proposed model’s relative efficiency with other baseline models. The results showed that our proposed model (AR + News) could significantly improve the forecasting accuracy, compared with other baseline models. Specifically, our proposed model outperformed the traditional influenza surveillance model based on historical official ILI rates (AR, *t* = −3.164, *p* = 0.002 in the south; *t* = −4.196, *p* = 0.000 in the north) and also outperformed the forecasting model incorporating microblog data as an exogenous input (AR + Mblog, *t* = −2.212, *p* = 0.029 in the south; *t* = −5.721, *p* = 0.000 in the north). When adding microblog data into our proposed model, the forecasting accuracy in RMSE decreased in both the south and north of mainland China. In the south, the accuracy decrease was not statistically significant (*t* = −1.047, *p* = 0.297), while in the north, the accuracy decrease was statistically significant at the level of 0.1 (*t* = −1.878, *p* = 0.063).

## 4. Discussion

Our proposed model innovatively incorporated online news articles as a new data source for influenza activity forecasting in both the south and north of mainland China. The results showed that online news articles could significantly improve the forecasting accuracy of influenza activity during outbreaks, compared to other online data sources. This accuracy improvement by mining online news articles could support the public health agency in making better preparations for influenza outbreaks, including improving resource allocation efficiency. This implication is even more important in a geographically vast and densely populated country like China.

Traditional influenza surveillance data from the CNIC is based on ILI reports from sentinel hospitals all over the country, which suffers from a 1–2 week delay for the processing and aggregating all the sentinel hospitals’ reports [5,7]. Online news articles, consisting of both user-generated news and organization-generated news [24], can help capture real-time influenza activity. Our results showed that the fraction of influenza-related online news articles peaked one week earlier than the official reports, and the two-week time lag for online news articles led to the highest prediction accuracy in the auto-regression model. This implies that online news articles, as a new online data source, could produce estimation results 1–2 weeks ahead of official ILI surveillance reports.

According to our analysis, about 93.0% of online news articles are from unofficial user-generated news, allowing users to post events happening around themselves in time. This character could help to capture a real-time snapshot of influenza-related activities. A large part of user-generated news comes from Wechat Official Account in China, where users can post news articles freely to attract clicks and generate income through advertisements. Similar characters have been found in other worldwide platforms like LinkedIn Articles, Medium, Instant Articles [47], etc., which shows that online news articles have the potential to enhance forecasting accuracy on influenza epidemics all over the world. 

Other online data sources, such as search queries, Twitter, or Microblogs, have shown great potential to supplement influenza detection. However, they also suffered from instability and linguistic noise [8,22], especially during influenza-like illness outbreaks [5,22]. Our study’s data is from 6 October 2019 to 17 May 2020, during which a pandemic of COVID-19 also broke out in January 2020 in China. COVID-19 has many similar symptoms compared to influenza, causing additional noise in online data sources. In our study, the forecasting accuracy decreased in the south of mainland China after incorporating microblog data into historical ILI rates. The decrease may be because of the noise produced by COVID-19. Meanwhile, online news articles contribute to a significant improvement in forecasting accuracy in both the south and north of mainland China. This proves that online news articles have a great potential to enhance influenza epidemics forecasting accuracy and robustness, especially during influenza-like illness outbreaks.

Our study also proposed a keyword-based method to extract influenza-related online news articles. This method framework could also help future research monitoring other diseases based on online news articles.

Several limitations and future research directions should be noted. First, only one flu season’s data was used in our analysis, and COVID-19 happening during this period could have had a confounding factor during the analysis. We could not extract additional data from the SNOSS, which only provides data within the most recent one and a half years. Even though the results from the south and north of mainland China both supported our arguments, future studies may test the methods in different years’ flu seasons for robustness.

Second, this study used the keywords-based method to extract influenza-related online news articles. We used this method because we wanted to explore the whole dataset of online news articles from the SNOSS. The original raw data could not be downloaded from SNOSS, and many other advanced machine learning methods could not be performed. Future research can try to test the model on actual raw data of official and unofficial online news articles and make use of other machine learning methods. More accurate data filtering processes and deeper content analyses can then be conducted, e.g., using classification algorithms to improve accuracy in filtering results [38] or using topic modeling algorithms to extract richer information from online news articles [48].

Finally, we did not compare our model with methods based on search query data, such as Google Flu Trends [7], in our research. Google Flu Trends stopped providing service on 9 August 2015. Even though people can still get flu-related Google search query data, Google Flu Trends has been shown to suffer from instability during influenza outbreaks [22]. However, future research could still add a comparison of search query data, if possible, to show a clearer picture of the difference between online news articles and other online data sources.

We would also like to discuss the public health implications of our findings in developing countries, including China. About 940 million internet users, 67.0% of the overall population, read and generate information online in China [49]. Among these users, 99.2% use a mobile phone to access the Internet and 30.4% are from rural areas. This character allows internet-based sources such as online news articles to monitor influenza activity in rural areas, where there are no sentinel hospitals to monitor influenza-related activities. Innovative approaches are also needed to monitor vulnerable subgroups who do not even have access to the Internet. Compared with other internet-based sources, online news articles contain richer information, helping extract more detailed disease characteristics and contribute to better preparation. As more and more online news articles are generated, it would be possible to detect various disease outbreaks in more specific areas and subgroups.

## 5. Conclusions

This study showed that unofficial online news articles accounted for the major part of present online news articles. Online news articles from both official and unofficial sources may serve as an innovative and effective exogenous input to supplement the traditional influenza surveillance model based on historical ILI data. Our proposed autoregression model based on online news articles significantly improved influenza forecasting accuracy during outbreaks, compared to the autoregression model based on historical ILI rates or additional microblogs. Our findings also revealed that online news articles could produce forecasting results 1–2 weeks ahead of official ILI surveillance reports.

## Figures and Tables

**Figure 1 ijerph-18-06591-f001:**
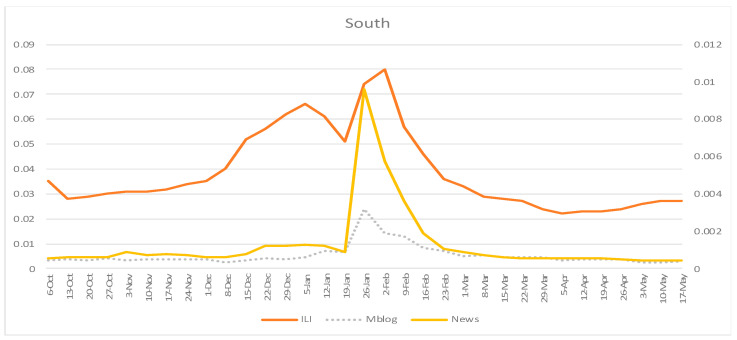
Weekly time series of ILI rates from the Chinese National Influenza Center (CNIC), fraction of influenza-related online news articles, and fraction of influenza-related microblogs: South of mainland China, 6 October 2019–17 May 2020. ILI = influenza-like illness rates from CNIC; Mblog = fraction of influenza-related microblogs; News = fraction of influenza-related online news articles.

**Figure 2 ijerph-18-06591-f002:**
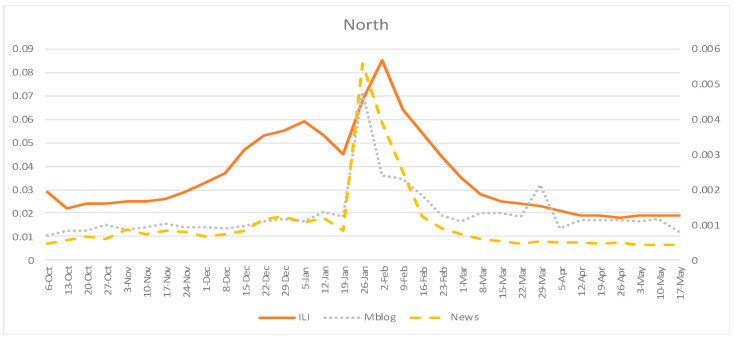
Weekly time series of CNIC’s ILI rates, fraction of influenza-related online news articles, and fraction of influenza-related microblogs: north of mainland China, 6 October 2019–17 May 2020. ILI = influenza-like illness rates from CNIC; Mblog = fraction of influenza-related microblogs; News = fraction of influenza-related online news articles.

**Figure 3 ijerph-18-06591-f003:**
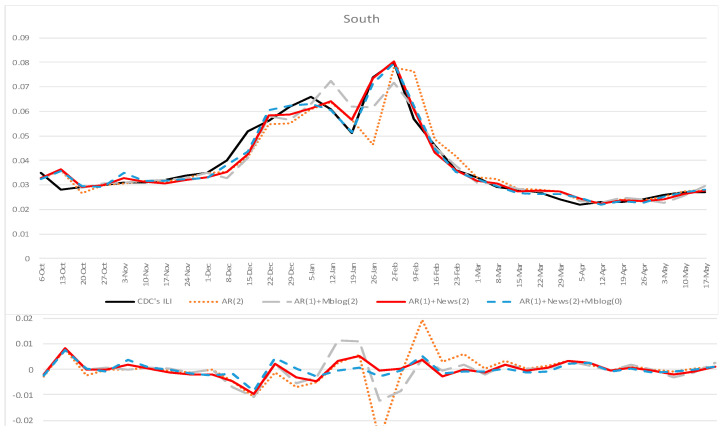
Forecasting results for the south of mainland China, 6 October 2019–17 May 2020. The weekly estimations of AR(2) (dotted orange), AR(1) + Mblog(2) (long dashed gray), AR(1) + News(2) (red), and AR(1) + News(2) + Mblog(1) (dashed blue) models were compared against the CNIC’s ILI rates (black). Bottom: The estimation error, defined as the estimated value minus the CNIC’s ILI rates.

**Figure 4 ijerph-18-06591-f004:**
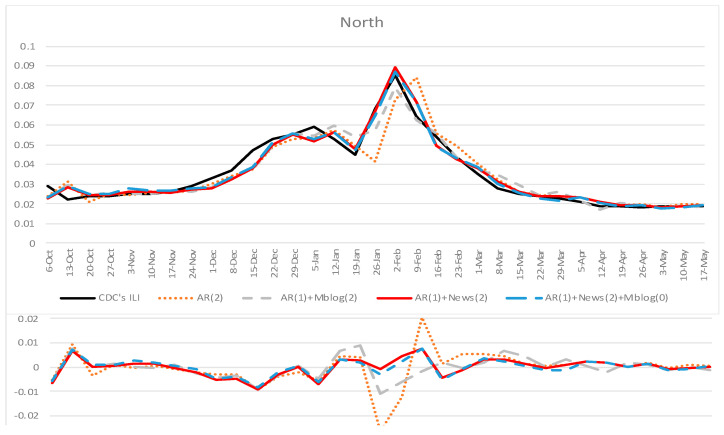
Forecasting results for the north of mainland China, 6 October 2019–17 May 2020. The weekly estimations of AR(2) (dotted orange), AR(1) + Mblog(2) (long dashed gray), AR(1) + News(2) (red), and AR(1) + News(2) + Mblog(1) (dashed blue) models were compared against the CNIC’s ILI rates (black). Bottom: The estimation error, defined as the estimated value minus the CNIC’s ILI rates.

**Table 1 ijerph-18-06591-t001:** Comparison of different models for the forecasting of influenza epidemics: south of mainland China.

Model (lag)	RMSE	R^2^	MAE	Correlation	Correlation of Increment	*t* (sig.)
AR(1) + News(2)	**0.087**	**0.938**	**0.072**	0.980	0.824	
AR(2)	0.126	0.918	0.098	0.903	0.286	−3.164 *** (0.002)
AR(1) + Mblog(2)	0.150	0.872	0.113	0.946	0.517	−2.212 ** (0.029)
AR(1) + News(2) + Mblog(0)	0.107	0.905	0.083	**0.985**	**0.839**	−1.047 (0.297)

RMSE = root-mean-squared error; R^2^ = coefficient of determination; MAE = mean absolute error. Values in bold highlight the best performance for each index in the south of mainland China, *** means significant at 1% level, and ** means significant at 5% level.

**Table 2 ijerph-18-06591-t002:** Comparison of different models for the forecasting of influenza epidemics: north of mainland China.

Model (lag)	RMSE	R^2^	MAE	Correlation	Correlation of Increment	*t* (sig.)
AR(1) + News(2)	**0.119**	**0.943**	**0.099**	0.978	**0.834**	
AR(2)	0.151	0.898	0.122	0.910	0.317	−4.196 *** (0.000)
AR(1) + Mblog(2)	0.142	0.908	0.116	0.967	0.703	−5.721 *** (0.000)
AR(1) + News(2) + Mblog(0)	0.129	0.922	0.107	**0.980**	0.820	−1.878 * (0.063)

RMSE = root-mean-squared error; R^2^ = coefficient of determination; MAE = mean absolute error. Values in bold highlight the best performance for each index in the north of mainland China, *** means significant at 1% level, and * means significant at 10% level.

## Data Availability

The data that support the findings of this study are available from the corresponding author, upon reasonable request.

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
