# Peer review of "Enhancing Influenza Epidemics Forecasting Accuracy in China with Both Official and Unofficial Online News Articles, 2019–2020"

_ijerph, 2021, doi:10.3390/ijerph18126591_

Round 1
Reviewer 1 Report
The authors developed a forecast model by examining both official and unofficial online news articles. Their model significantly improved the forecasting accuracy, compared to other influenza surveillance models based on historical ILI rates or adding microblog data as an exogenous input. Also, they found that influenza forecasting based on online news articles could be 1-2 weeks ahead of official ILI surveillance reports. This is a very meaningful work.
A few concerns are that:
- I noticed that the author only used the data between October 6, 2019 and May 17, 2020, which is defined as the regular flu season. Did the author tested their model using the data that is not fall in the flu seasons? what will the results be like? What will the differences be there between applying model to flu season and non-flu season?
- The author used the data from a commercial platform, so the data may have already been preprocessed by the platform, so this may provide a bias for the model that can be easily developed for the forecast, did the author test their model using the actual raw data of unofficial articles.
Reviewer 2 Report
Very interesting research and promising results. However,
- The glossary of influenza (flu) terms from the CDC is English Version. How did you use it for Chinese text processing? Where is table S1? later S2?
- The authors used many different methods with references. But they are hard to follow in this way.
- From common sense, forecasting based on other online resources can also be 1-2 weeks ahead of official ILI surveillance reports. It is more important to show how accuracy can be improved by mining news articles.
- English is fine. Sometimes, the authors use both past tense and present tense in one paragraph. It is better to use only past tense or present tense in these situations.
Reviewer 3 Report
The paper proposed a new model based on online news articles for improving the influenza forecasting accuracy during outbreaks. The authors present their motivation with clarity, the work is well-structured and written. Moreover, the results are solid and very interesting. In this line, the discussion section covers the main limitations and future work.
I enjoyed reading this paper and look forward to seeing it published.
Author Response
Thank you very much for your encouraging words!
Reviewer 4 Report
In the paper, the authors present an auto-regressive approach based on official ILI rates, the fraction of flu-related online news articles, and the current week's microblog fraction to predict the logit-transformed ILI weekly rate. Although this paper does not propose novel technical improvements from the current literature on this domain, the author adopted the existing methods to claim new improvements in forecasting accuracy and producing earlier predictions.
Some problems in the paper need to be fixed:
A. One of the major aspects that you already pointed out in the discussion (section 4) is the lack of performance comparison with existing influenza forecasting (or broader scope disease forecasting) methods. There are considerable previous studies in this domain, especially machine learning models. I am curious about the performance of your proposed method compared to such machine learning models.
B. I think the literature review is not completed. Please do not focus on influenza forecasting studies but also in broader terms – disease prediction. Please look at the following related works and others:
- Poirier, C., Liu, D., Clemente, L., Ding, X., Chinazzi, M., Davis, J., ... & Santillana, M. (2020). Real-time forecasting of the COVID-19 outbreak in Chinese provinces: machine learning approach using novel digital data and estimates from mechanistic models. Journal of medical Internet research, 22(8), e20285.
- Gupta, A., & Katarya, R. (2020). Social media based surveillance systems for healthcare using machine learning: A systematic review. Journal of Biomedical Informatics, 103500.
- Allam, Z., Dey, G., & Jones, D. S. (2020). Artificial intelligence (AI) provided early detection of the coronavirus (COVID-19) in China and will influence future Urban health policy internationally. AI, 1(2), 156-165.
- Rees, E. E., Ng, V., Gachon, P., Mawudeku, A., McKenney, D., Pedlar, J., ... & Knox, J. (2019). Early detection and prediction of infectious disease outbreaks. CCDR, 45, 5.
- He, G., Chen, Y., Chen, B., Wang, H., Shen, L., Liu, L., ... & Min, Z. (2018). Using the Baidu search index to predict the incidence of HIV/AIDS in China. Scientific reports, 8(1), 1-10.
- Yan, S. J., Chughtai, A. A., & Macintyre, C. R. (2017). Utility and potential of rapid epidemic intelligence from internet-based sources. International Journal of Infectious Diseases, 63, 77-87.
- Nsoesie, E. O., & Brownstein, J. S. (2015). Computational approaches to influenza surveillance: beyond timeliness. Cell host & microbe, 17(3), 275-278.
- Bernardo, T. M., Rajic, A., Young, I., Robiadek, K., Pham, M. T., & Funk, J. A. (2013). Scoping review on search queries and social media for disease surveillance: a chronology of innovation. Journal of medical Internet research, 15(7), e147.
- Wilson, K., & Brownstein, J. S. (2009). Early detection of disease outbreaks using the Internet. Cmaj, 180(8), 829-831.
C. Both Figures 1 and 2 are quite small, so it is difficult for me to view them (I have to zoom in 300% to read the labels). Please split them into smaller figures, i.e., each sub-image for North and South.
D. At lines 211 and 222, you should not use "ref. xx". You need to use the citation format [16], [24].
E. Missing the article "the" before "south and north of mainland China" (check line 151, 252), "repeated k-fold cross-validation approach" (line 215).
F. I think the following terms should be capitalized: Twitter, Microblog (check line 315), TF-IDF (check line 116)
G. You haven't explained the last column in Tables 1 and 2. Why did you put "***", "**", "*" after some values in this column? And, what is the meaning of the value outside and inside the parentheses in this column? Although I assume you display the t statistic test results, they may not know about them for many readers.
H. There are some long sentences in section 3, especially in the last two paragraphs. Please consider rewriting them in smaller sentences.
Round 2
Reviewer 4 Report
I am happy to support the publication of this manuscript after carefully checking the revised version, which has covered most of my comments on the previous one.